# Bi-enzymatic chemo-mechanical feedback loop for continuous self-sustained actuation of conducting polymers

Serena Arnaboldi [1,4], Gerardo Salinas [2,4], Sabrina Bichon[3], Sebastien Gounel [3], Nicolas Mano [3] & Alexander Kuhn [2] ✉

Artificial actuators have been extensively studied due to their wide range of applications from soft robotics to biomedicine. Herein we introduce an autonomous bi-enzymatic system where reversible motion is triggered by the spontaneous oxidation and reduction of glucose and oxygen, respectively. This chemo-mechanical actuation is completely autonomous and does not require any external trigger to induce self-sustained motion. The device takes advantage of the asymmetric uptake and release of ions on the anisotropic surface of a conducting polymer strip, occurring during the operation of the enzymes glucose oxidase and bilirubin oxidase immobilized on its surface. Both enzymes are connected via a redox polymer at each extremity of the strip, but at the opposite faces of the polymer film. The time-asymmetric consumption of both fuels by the enzymatic reactions produces a double break of symmetry of the film, leading to autonomous actuation. An additional break of symmetry, introduced by the irreversible overoxidation of one extremity of the polymer film, leads to a crawling-type motion of the free-standing polymer film. These reactions occur in a virtually unlimited continuous loop, causing long-term autonomous actuation of the device.

Artificial muscles, based on dielectric elastomers[1], carbon nanotubes[2,3], artificial proteins[4,5] and conductive textiles[6,7], have gained considerable attention due to their interesting potential applications in soft robotics, nanotechnology and biomedicine[8–14]. This type of actuation is commonly driven by external electrical, chemical, light or thermal stimuli[15–19]. In this context, an interesting option is the use of conducting polymers (CPs)[20,21], as artificial actuators, since their motion can be triggered at low voltages, comparable to those present in mammalian muscles. CPs present considerable advantages in comparison with other materials, due to the easy fine-tuning of their volume, color, wettability, and surface conformation during the oxidation/reduction process[22–24]. In particular, volumetric variations occur as a result of an electrochemically induced ion exchange[25,26], accompanied by conformational changes of the polymeric chains,

producing an electromechanical deformation[27–29]. Actuators based on CPs can accommodate higher stress and strain in comparison to natural muscles, allowing the use of these materials to develop artificial muscle-type devices[30]. Among the various existing CPs, polypyrrole doped with dodecylbenzenesulfonate (Ppy/DBS) exhibits one of the highest actuation efficiencies, originating from the large volume change that occurs during redox cycling, in conjunction with a high mechanical strength[31–33]. However, in contrast to natural muscles, actuators based on CPs require usually a direct physical connection to an electric power supply in order to trigger the electromechanical deformation[34], which in many cases severely limits the range of applications. An alternative approach is the wireless operation of such devices, either by the use of oxidizing/reducing agents[35,36] or by causing an asymmetric polarization of the material. The latter can be

[1]Dip. Di Chimica, Univ. degli Studi di Milano, Milan, Italy. [2]University Bordeaux, CNRS, Bordeaux INP, ISM, UMR 5255, F-33607 Pessac, France. [3]Centre de Recherche Paul Pascal, University Bordeaux, CNRS, UMR 5031, Pessac, France. [4]These authors contributed equally: Serena Arnaboldi, Gerardo Salinas. ✉e-mail: kuhn@enscbp.fr

achieved by an external electric field, as in the case of bipolar electrochemistry[37–39], or by coupling thermodynamically spontaneous reactions to each extremity of the device[40]. Although bipolar electrochemistry has been used successfully to produce electromechanical deformation of conducting polymers as transduction signal of chemical information in solution[41–45], it can't be considered as a completely autonomous system because an external electric field is needed as a driving force.

In this context, enzymatic reactions are an elegant solution, providing the necessary driving force, allowing the spontaneous conversion of biochemical energy into motion[46,47]. Recently, a biocatalytic liquid crystal elastomer that responds to the presence of urea has been proposed, based on the covalent immobilization of urease in the responsive host matrix. This device undergoes shrinking due to the formation of ammonia, caused by the enzymatic reaction, which triggers an irreversible collapse of the network, leading to a decrease of the liquid crystalline order inside the film[48]. However, the applicability of this system as an artificial muscle is limited by the irreversible shrinking process inside the elastomer matrix. An interesting alternative is to use not only one, but two coupled enzymatic reactions to power reversibly the spontaneous actuation of different devices[49]. For example, when integrated in bioelectrodes, the power output of biofuel cells can drive microdevices that require a relatively low driving force[50]. A combination of enzymatically driven bioelectrodes that convert glucose and oxygen are especially interesting as they allow the spontaneous conversion of chemical energy present in body fluids[51]. Mano et al. reported the bioelectrochemical propulsion of carbon fiber swimmers, by coupling the oxidation of glucose with the reduction of oxygen, driven by glucose oxidase (GOx) and bilirubin oxidase (BOD), respectively[52]. Strack et al. used enzymatic reactions in solution to design a logic circuit controlling a biochemical actuator[53]. Despite these interesting findings, the design of a completely autonomous artificial muscle, showing reversible motion loops driven by biochemical reactions, remains still an important open challenge.

Herein we present a soft bioelectrochemical actuator based on a conducting polymer support, where self-sustained periodic motion is triggered by the spontaneous enzymatic oxidation and reduction of glucose and oxygen. The device takes advantage of two main ingredients; (1) the asymmetric uptake and release of ions on the anisotropic surface of a PPy film and (2) the spontaneous enzymatic oxidation and reduction of glucose and oxygen, driven by GOx and BOD, respectively. The synergy of these two ingredients allows designing an original system, where the enzymatic reactions provide enough driving force to trigger periodic deformation via a redox feedback loop. The spatially separated time-asymmetric consumption of both fuels by the enzymatic reactions, produces a double break of symmetry leading to an efficient long-term autonomous actuation.

## Results

### Design of the bioelectrochemical actuation loop

The bioelectrochemical actuator was designed by immobilizing separately the corresponding enzymes, together with the appropriate osmium-based redox polymers as mediators, ensuring the electrochemical wiring of the enzymes[54,55], at each extremity and at opposite faces of an oxidized Ppy/DBS strip (1 cm×1.5 cm) (Fig. 1 red square). The polymer strip has been generated by galvanostatic electropolymerization on a gold-coated glass slide and was then peeled off to get a freestanding film. The potential transient curve obtained during the electropolymerization exhibits a constant potential value in a range between 0.65 and 0.7 V vs Ag/AgCl, below the overoxidation potential values of the monomer and the polymer (Supplementary Fig. 1). Since the pristine and rather thick Ppy film ($\approx 70\ \mu m$) is, after its generation by electropolymerization, in an oxidized state, and the rather big DBS anions remain trapped inside the polymer matrix, a reversible volume change can occur based on an uptake or release of

cations, i.e., $Na^+$ or $K^+$, when its redox state is changed. Considering that the efficiency of the actuation depends strongly on the size of the hydration shell of the cation, the volume change associated with the redox process is caused mainly by an exchange of $Na^+$ due to its larger hydration shell in comparison with $K^+$ ($\approx 4.4$ and $\approx 2.1$, respectively)[56,57]. In addition, the electropolymerized film presents an anisotropic surface, with one face having a rough morphology (porous), whereas the opposite face being smooth and more compact (Supplementary Fig. 2). This feature enables an asymmetric uptake/release of ions, which translates into a pronounced bending. The anodic extremity of a PPy strip was generated by coating its rough face with a GOx hydrogel, containing an osmium redox polymer having the appropriate redox potential, and a cross-linker. A hydrogel of similar composition, but with an osmium redox polymer having a more positive redox potential, was used to immobilize BOD at the other extremity of the PPy strip and on the smooth face of the film, constituting in this way the cathodic part of the redox loop. The direct conversion of bioelectrochemical energy into electromechanical motion was achieved by placing the modified freestanding Ppy/DBS strip at the bottom of a cell containing an $O_2$-saturated phosphate buffer solution (0.3 M PBS, pH = 5 at 25 °C). In the presence of glucose, oxidation is triggered spontaneously by GOx at the anodic side of the device (Fig. 1A). This leads to an electron transfer to the oxidized Ppy, triggering its reduction. Due to the asymmetric incorporation of cations into the polymer matrix through the rough face of the strip, inducing a local swelling, the film bends downwards at its extremities, thus lifting the center part of the film (Fig. 1B). This deformation allows now a more efficient diffusion of oxygen towards the location where BOD is immobilized, thus enabling the electrons, which are temporarily stored in the conducting polymer, to be transferred to this enzyme for the reduction of oxygen to water (Fig. 1C). This local consumption of electrons, accompanied by the asymmetric release of cations from the polymer matrix, again at the rough interface, brings the conducting polymer back to its initial redox state, and consequently also cancels the associated bending (Fig. 1D). The restoration of the flat configuration of the polymer strip consequently shuts down the oxygen supply to BOD. This sequence of electron transfer reactions, together with the associated deformations, is then repeated and constitutes an electromechanical feedback loop, leading to a periodic autonomous oscillation of the polymer shape, as long as both enzyme substrates are present.

Theoretically, all the redox processes that occur during the entire feedback loop are spontaneous from a thermodynamic point of view (red arrow, Fig. 1E). The two main reactions, at the extremities of Fig. 1E, are the oxidation of glucose ($Glu_{red}$) into gluconolactone ($Glu_{ox}$) and the reduction of $O_2$ into $H_2O$, enabled by GOx and BOD, respectively. The energy difference between these two enzymatic reactions ($\Delta G = -223\ kJ/mol$) allows a global downhill flow of electrons, mediated by appropriate redox polymers, comparable to what happens in biofuel cells, and transiting via the Ppy film. The oxidation of glucose by GOx induces the reduction of the anodic redox polymer (from $Os^{3+}$ to $Os^{2+}$), and the $Os^{2+}$ in turn transfers an electron to the polymer strip, leading to its spontaneous reduction (from $Ppy^{(n+1)}$ to $Ppy^{n+}$). In the following step, these electrons are used for the reduction of the cathodic redox polymer (from $Os^{2+}$ to $Os^+$). Thus the Ppy film acts as an intermediate electron reservoir, due to the oxidation of the $Ppy^{n+}$ back to $Ppy^{(n+1)}$. Finally, the spontaneous oxidation of $Os^+$ back to $Os^{2+}$ mediates the electron transfer to BOD, causing the reduction of oxygen. In theory, during these different redox processes, cations are entrapped and released by Ppy in order to guarantee electroneutrality, causing the swelling and shrinking of the system (Fig. 1E, gray region).

### Spontaneous bioelectrochemical actuation

In order to investigate in more detail, the thermodynamics of the reactions associated with the spontaneous feedback loop, both, the anodic and the cathodic sides, were first characterized separately by

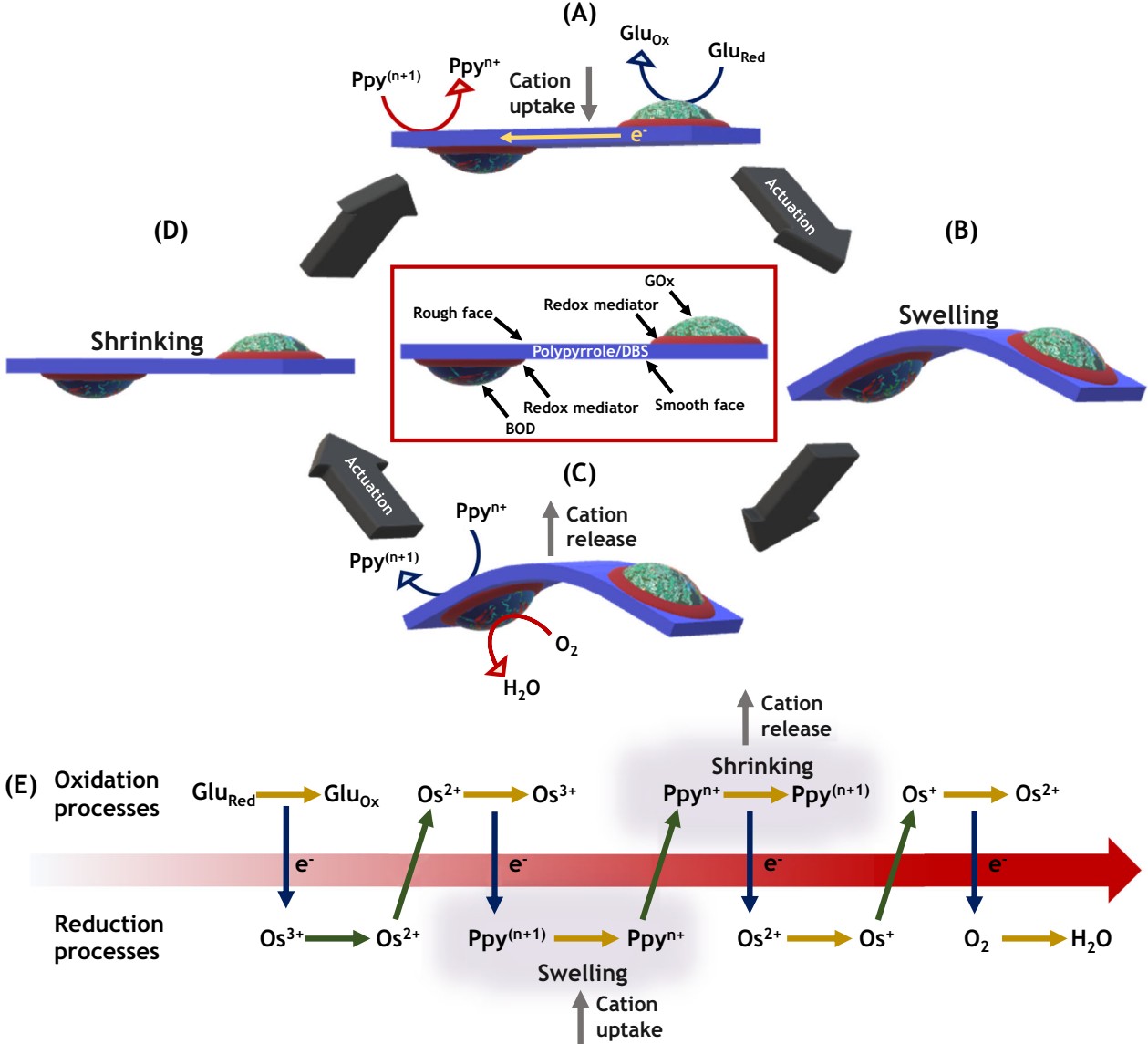

**Fig. 1 | Design of the bioelectrochemical actuation loop. A–D** Illustration of the actuation mechanism with a representation of the associated reactions. The red square inset shows the side view of the bioelectrochemical actuator, with the red part symbolizing the two different osmium-based redox polymers and the cross-linker, whereas the blue color indicates the Ppy/DBS film. **E** Schematic of the sequence of redox reactions involved in the autonomous loop. The gray region stands for the Ppy swelling and shrinking process. The arrow in the figure illustrates that the coupled redox reactions follow a thermodynamic chain which is an overall downhill reaction.

cyclic voltammetry under argon at a scan rate of $10\ mV\ s^{-1}$ in a 0.3 M PBS solution (pH = 5). The potentiodynamic measurements were carried out at 37.5 °C, since this is the optimum temperature where the enzymes exhibit the highest activity. The electrodes were prepared by immobilizing, separately, each enzyme with its corresponding redox mediator on the surface of a free-standing Ppy/DBS film, acting in this case as a conventional conductor. As it can be seen in Fig. 2A, in the absence of substrate (glucose or oxygen) only two redox peaks, corresponding to the reversible electrochemical oxidation/reduction of the osmium redox centers, can be observed at +0.05 V and +0.3 V vs Ag/AgCl, for the anodic (red line) and cathodic (blue line) osmium redox polymer, respectively. In addition, the voltammetric curve of the pristine Ppy film shows a reversible electroactive window between −0.20 V to 0.8 V vs Ag/AgCl. Such a redox behavior allows us to assume that Ppy is able to exchange electrons with both redox polymers, thus acting as an intermediate electron reservoir. These results confirm the thermodynamic viability of the sequence of redox reactions, enabling

the electron transfer from the anodic to the cathodic extremity of the Ppy strip. As can be seen, both enzymatic redox reactions occur in a potential region where Ppy is electroactive, avoiding any over-oxidation of the polymer matrix (Fig. 2A orange region). In fact, in this potential interval, between 0.0 V and 0.5 V vs Ag/AgCl, it is possible to assume that Ppy essentially acts as a conductor, leading to a capacitive current associated with the Ppy film.

To study the two involved individual spontaneous enzymatic redox reactions and the associated polymer deformation, two additional experiments were carried out. Two separate Ppy/DBS strips were fixed on a support with the rough face oriented upwards, leaving a fraction of the Ppy strip as a free-standing cantilever (Supplementary Fig. 3). In order to simulate the charge-transfer between the enzymatic system and the conducting polymer and its consequences on the shape of the polymer strip, two different initial redox states of the Ppy film were used to generate a reversible upward and downward bending. An oxidized film with immobilized GOx/redox mediator, and a

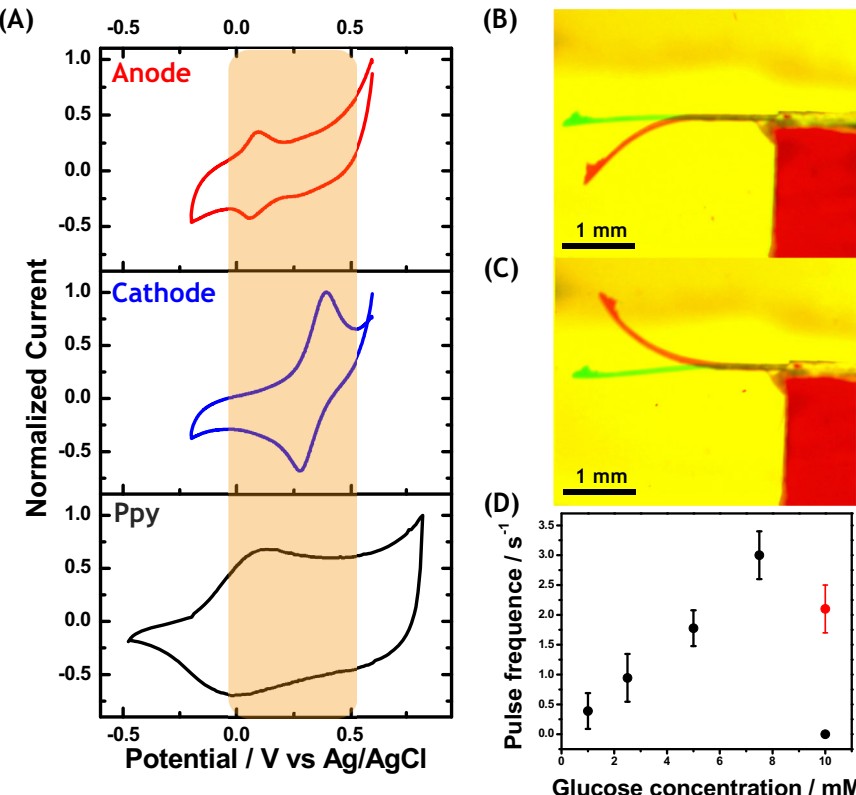

**Fig. 2 | Thermodynamic characterization, autonomous mono-enzymatic actuation and quantitative analysis of the oscillation frequency of the full bi-enzymatic devices. A** Cyclic voltammograms of a free-standing Ppy strip modified with GOx/Os-polymer/PEGDGE (red curve) and BOD/Os-polymer/PEGDGE (blue curve) recorded at 10 mV s⁻¹ at 37.5 °C, in argon saturated 0.3 M PBS buffer solution (pH = 5) in the absence of glucose and oxygen. For comparison, the black line indicates the cyclic voltammogram of a polypyrrole film, recorded in a 0.1 M TBAP/ACN solution at 10 mV s⁻¹. The orange region represents the potential region where Ppy is electroactive and stable. Autonomous bioelectrochemical

actuation of (**B**) a Ppy/DBS/GOx film in a 0.3 M PBS (pH = 5), 1 mM glucose solution and (**C**) a Ppy/DBS/BOD film in a 0.3 M PBS solution at 25 °C (pH = 5) saturated with O₂. The green line indicates the initial position of the actuator, whereas the red one corresponds to the final state. **D** Oscillation frequency values as a function of glucose concentration for the bioelectrochemical actuation of modified bi-enzymatic Ppy/DBS films in a 0.3 M PBS solution at 25 °C (pH = 5) saturated with O₂. The experiment represented by the red dot has been performed in naturally aerated solution, but with a twice-higher loading of BOD redox hydrogel. Data are presented as the mean values of three repetitions (mean ± 0.04).

partially reduced film with BOD/redox mediator were prepared. In a 0.3 M PBS (pH = 5), 1 mM glucose solution, the oxidized Ppy/DBS/GOx film shows a downward bending (Fig. 2B, Supplementary Movie 1, left side). Under these conditions, the GOx-driven spontaneous oxidation of glucose causes the reduction of the partially oxidized Ppy/DBS film (Supplementary Fig. 3A). This leads to the asymmetric uptake of cations through the rough interface, generating a swelling of the upper face of the film, causing a downward bending. On the contrary, an upward bending is observed for the Ppy/DBS/BOD film in a 0.3 M PBS solution (pH = 5) saturated with O₂, (Fig. 2C, Supplementary Fig. 3B, Supplementary Movie 1, right side). In this case, the enzymatic reduction of oxygen leads to the oxidation of the partially reduced Ppy/DBS film. This induces an asymmetric release of cations and shrinking at the upper rough face of the film and thus an upward bending of the cantilever. It is important to highlight that the dynamic behavior analysis was carried out at 25 °C in order to avoid thermal convection, which can influence the motion.

## Self-sustained enzymatic electromechanical motion
After this set of mono-enzymatic experiments, the spontaneous deformation of the full bi-enzymatic, not immobilized actuator, was studied in a 0.3 M PBS (pH = 5), 1 mM glucose solution saturated with O₂. Under these conditions, the device presents a continuous and periodic bending/stretching process, with an oscillation frequency of 0.45 s⁻¹ (Fig. 2D, Supplementary Movie 2, left frame). Moreover, a correlation between the frequency and the glucose concentration was

obtained, reaching up to 3 s⁻¹ for 7.5 mM glucose (Fig. 2D, Supplementary Movie 2, frame 2 to 4). However, at even higher glucose concentrations (10 mM), the biochemical actuator remains in a fully bent steady state (Supplementary Movie 2, frame 5). This indicates that, in an O₂ saturated solution, the overall electron transfer kinetics across the whole reaction chain is limited by the enzymatic oxidation of glucose when low glucose concentrations are used (<7.5 mM), transduced by a continuous change in oscillation frequency. In contrast to this, for glucose concentrations above 7.5 mM, the continuous oxidation of glucose at the anodic extremity continuously provides electrons at a rate that exceeds the flow of electrons that can be handled by the oxygen reduction at the cathodic side, thus leading to a constantly bent shape. In order to confirm this hypothesis, an additional experiment has been carried out for which the amount of immobilized BOD has been doubled so that the conversion of oxygen becomes faster. In this case, oscillations reappear for the same high glucose concentration of 10 mM, even if the solution is only saturated with air instead of pure oxygen (Fig. 2D, red dot). It is important to highlight that the continuous self-sustained bending/stretching process remains operational for at least ten minutes. For longer times, a slow decay of the actuation can be observed, which is associated mainly with two factors, leakage of the enzyme from the surface of the device, and an intrinsic decay of enzymatic activity. However, both aspects might be optimized in future work, because it is possible to cover the enzyme layers with protecting membranes and thus avoiding the loss/leaking of enzymes due to the mechanical stress related to the

local hydrodynamics and shear forces present during actuation. Furthermore, it is possible to use genetically modified enzymes which have a longer lifetime than the enzymes used in this proof-of-concept study.

## Lateral motion of the bio-electromechanical actuation loop

In order to transform this periodic shape change into a directional motion, an additional break of symmetry is needed[58]. This can be easily achieved by an irreversible slight overoxidation of one extremity of the polymer film, which leads to a localized increase in stiffness. Such a stiffness should allow for generating a crawling-type motion, since the reversible bending/stretching process occurs now only at the end of the polymer strip which is still fully electroactive[59]. The asymmetric overoxidation was carried out by bipolar electrochemistry prior to the modification of the film with enzymes. The asymmetric polarization caused by bipolar electrochemistry leads to Ppy having three different spatially separated redox states; reduced Ppy at the $\delta^-$ extremity, oxidized Ppy in the middle part of the strip, and overoxidized Ppy at the $\delta^+$ end[60] (Supplementary Fig. 4). Such a local difference in electroactivity

allows actuation by following a slightly different bioelectrochemical reaction pathway (Fig. 3A–D). First, the enzymatic oxidation of glucose on the right side causes the reduction of oxidized Ppy in the middle section of the polymer strip, accompanied by an uptake of cations, and thus by a localized swelling and a relative downward bending of the two extremities of the film. Subsequently, this gives BOD access to oxygen, and the enzymatic oxygen reduction withdraws electrons from the reduced Ppy film, leading to the corresponding upward bending. This closes the reaction feedback loop and generates, as previously, a shape oscillation. However, it is important to point out that in this case, both enzymes are drop cast in a section of the asymmetrically modified Ppy strip where the transfer of electrons is possible, thus closer to the middle part of the film. As can be seen, the additional asymmetry, induced by the stiff section of the polymer, enables a continuous motion, translating into a lateral displacement (Fig. 3D, E and Supplementary Movie 3). The experiment was carried out by placing an asymmetrically modified bioelectrochemical actuator strip (0.5 cm × 1.5 cm) in a 0.3 M PBS (pH = 5), 5 mM glucose solution saturated with $O_2$. Under these conditions, the device presents a

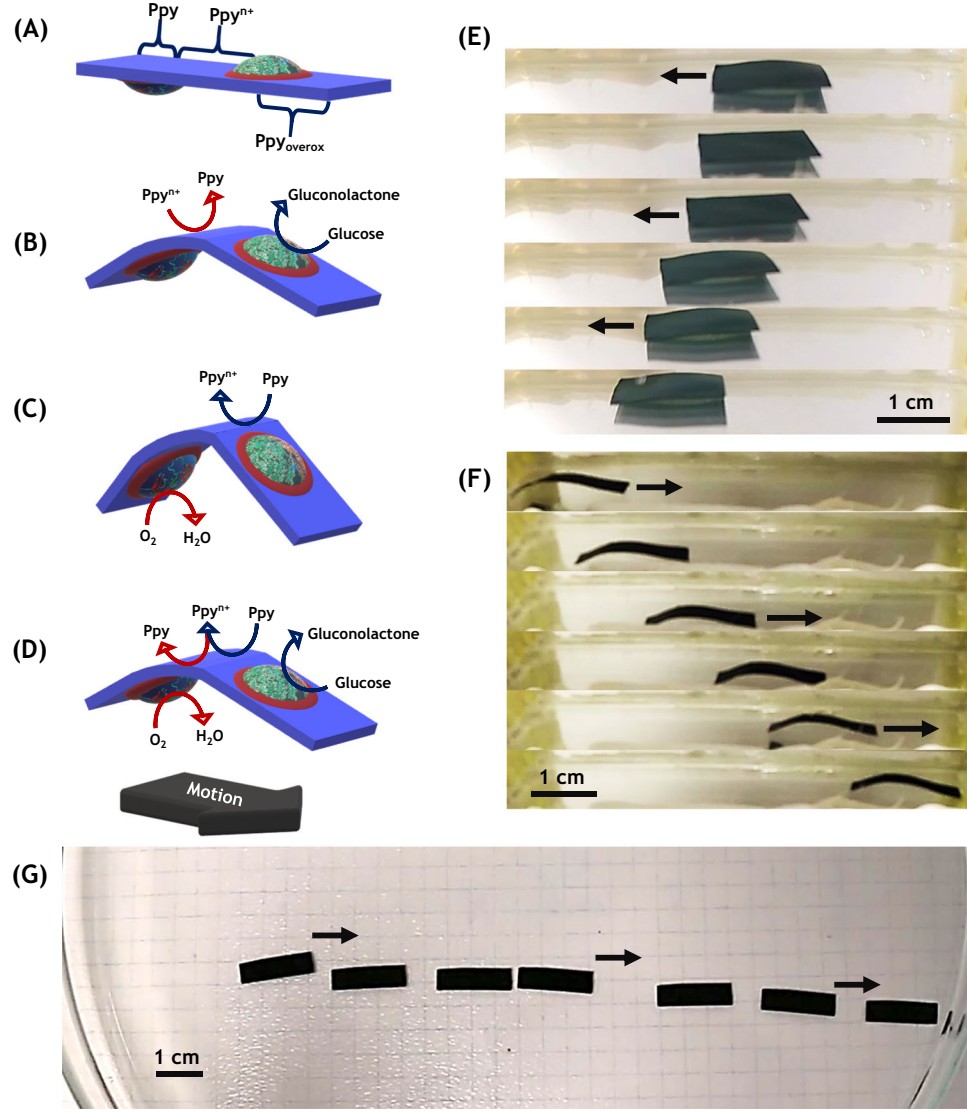

**Fig. 3 | Generation of autonomous lateral motion. A–D** Schematic illustration of the mechanism generating a global lateral motion. The red part symbolizes the redox polymers and cross-linker, whereas the blue color represents the Ppy/DBS film. **E** Macroscopic lateral motion of an asymmetrically modified Ppy/DBS film immersed in a 0.3 M PBS at 25 °C (pH = 5), 5 mM glucose solution, saturated with $O_2$ (see Supplementary Movie 3). **F** The same experiment as in (**E**), but using an actuator with a shorter region of electroactive Ppy and an inverted orientation (see Supplementary Movie 4). **G** Top view of a long distance motion under the same conditions as in (**E**) with an average speed of 1 mm/s (Supplementary Movie 6).

rather linear motion with an average speed of 0.8 mm s$^{-1}$, and the deformation takes essentially place in the middle section of the polymer film, which is the most electroactive part of the device. Since each redox state present in the film has different resistance values, in comparison with the pristine Ppy, resistance measurements before and after the bipolar electrochemical modification of the polymer were carried out in six different regions along the Ppy film. The pristine Ppy film presents a resistance value around 380 ohms (0.56 S cm$^{-1}$ Supplementary Fig. 5A). After 10 seconds of polarization in a bipolar electrochemical preconditioning experiment by applying an electric field of 4 V cm$^{-1}$, an increase of one order of magnitude in the resistance values at each extremity of the polymer film was observed, with a conducting region of approximately 0.9 cm width at the middle part of the strip (Supplementary Fig. 5B). As stated above, this is attributed to the reduction and overoxidation of the oxidized Ppy film[60]. By increasing the polarization time (20 seconds), the size of the regions with high resistance increases, and also the value of the resistance increases by two orders of magnitude. Consequently, the width of the zone with higher conductivity decreases (0.3 cm) (Supplementary Fig. 5C). Modulating the size of this latter region and the orientation of the actuator should allow for fine-tuning the motion. In order to test this hypothesis, the motion of an asymmetrically modified bioelectrochemical actuator with a smaller region of electroactive Ppy, and an orientation changed by 180°, was recorded in a 0.3 M PBS (pH = 5), 5 mM glucose solution saturated with O$_2$. Under these conditions, the device presents a motion with an average speed of 3.6 mm s$^{-1}$ (Fig. 3F, Supplementary Movie 4). As expected, the device now moves in the opposite direction, and the pronounced difference in speed between both experiments is caused by the difference in electroactive area available for the pulse mechanism.

For a smaller electroactive area, the local ion exchange current density is higher, leading to a spatially more confined swelling/shrinking and therefore to a more pronounced and more efficient bending. The motion is associated with an additional symmetry breaking, caused by a difference in the ion uptake and release kinetics at the rough face of the Ppy film during the charging/discharging process. In order to corroborate this hypothesis, double pulse potentiostatic measurements were carried out with a freestanding Ppy film in a 0.1 M LiClO$_4$ solution. The plot of the angular speed as a function of time shows that the ion release, during the oxidation of the film, is more than one order of magnitude faster than the ion uptake (Supplementary Fig. 6 and Supplementary Movie 5). This difference in ion exchange kinetics induces a continuous pulse process in the electroactive middle part of the polymer. Finally, in order to test the durability of the generated motion, the displacement of the hybrid device was tested over a longer distance. In this case, the experiment was carried out by placing an asymmetrically modified bioelectrochemical actuator strip (0.5 cm × 1.5 cm, wireless asymmetrical modification: 4 Vcm$^{-1}$ for 10 seconds) in a 0.3 M PBS (pH = 5), 5 mM glucose solution saturated with O$_2$. As it can be seen, under these conditions the hybrid actuator presents a characteristic pulse motion with an average speed of 1 mm s$^{-1}$, which is in good agreement with the speed obtained with other similar Ppy/DBS strips (Fig. 3G and Supplementary Movie 6). Since the Supplementary Movie 6 presents the top view of the motion, the vertical deformation of the polymer strip can't be easily seen and therefore motion appears to occur in a sequence of jumps.

## Discussion

A completely autonomous soft bioelectrochemical actuator, where motion is triggered by coupling spontaneous enzymatic reactions via an integrated feedback loop, was successfully designed. Such a device takes advantage of the synergy between the electromechanical properties of polypyrrole, associated with a time-asymmetric volume change, and the thermodynamic driving force provided by the coupled enzymatic oxidation and reduction of glucose and oxygen,

respectively. In this context, the mechanism of motion generation is based on a cascade-type sequence of thermodynamically spontaneous redox reactions, which leads to an overall energetic downhill flow of electrons from the anode to the cathode of the device. The combination of these electromechanical and bioelectrochemical processes produces a continuous periodic bending/stretching motion (pulse mechanism), with an oscillation frequency depending on the glucose concentration in solution. Moreover, due to the introduction of an additional symmetry break, leading to controlled variations of the electric resistance along the polymer film, it is possible to induce a directional motion of the device. In this case, the bioelectrochemical feedback scheme generates pulses of periodic mechanical deformation of the polymer strip, which are transduced into a global displacement, with a speed that can be controlled by the degree of asymmetry in terms of conductivity. These proof-of-concept experiments demonstrate that with actuators having a rather simple and straightforward design, it is possible to convert directly (bio)chemical energy into self-sustained periodic motion, without using materials that require complex synthetic routes or additional external stimuli[61–65]. Moreover, it is possible to assume that fine-tuning the thickness and conductivity of the Ppy films allows modulating the bending performance, since higher conductivity values facilitate the electron exchange from the anode to the cathode of the device, whereas thinner films favor the ion exchange within the polymer matrix. In a more general context, it can be envisioned that the proposed system, which can be considered as a short-circuited biofuel cell, can be extended to alternative combinations of enzymes, based on a downhill electron flow. One possible generalization of this concept could be, for example, the coupling of BOD with galactose oxidase (GalOx), since the latter can be used for the electro-oxidation of different alcoholic substrates into their corresponding aldehydes[66]. Nonetheless, the possible leakage of the enzyme from the electroactive surface or the control of the enzymatic activity are factors that need to be considered in order to extend the lifetime of the bioelectromechanical actuation.

In conclusion, such bioelectrochemical actuators, which represent the equivalent of short-circuited biofuel cells, have the potential to be used in biomimetic and soft robotic applications, since the here proposed approach is completely wireless and self-sustained. The enzymatic reactions can be modulated by changing the enzyme type according to the available biofuel. These reactions can take place in a continuous loop, causing long-term autonomous actuation. Polypyrrole is easily prepared by electropolymerization, exhibiting a large accessible electroactive surface area, excellent electron donor-acceptor interactions, high charge storage abilities, and fast ion motion within the polymer matrix. These systems are endowed with high actuation performances, including a large bending strain, with the capability to generate a linear gliding motion by breaking the symmetry through overoxidation. Therefore, the proposed system opens up interesting applications as multifunctional smart dynamic materials, truly mimicking natural muscles driven by physiologically available fuel.

## Methods

### Electrosynthesis of Ppy/DBS

A solution of pyrrole monomer (0.2 M, Sigma-Aldrich 98%) was dissolved in an aqueous sodium dodecylbenzenesulfonate (DBS) (0.25 M, Sigma-Aldrich, technical grade) solution. Two commercial gold-coated glass slides (Kerdry, HEF® photonics) were positioned parallel in a beaker filled with 12 cm$^3$ of this solution. These gold-coated glass slides were used as a working and counter electrode, whereas Ag/AgCl (3 M, KCl, Sigma-Aldrich ≥99.0%) was the reference electrode. The electropolymerization of pyrrole was carried out by applying a constant current of 4 mA for 1.5 h. After polymerization, the polypyrrole (Ppy) film was washed with water, dried, and peeled off from the substrate.

All aqueous solutions were prepared with deionized water (MilliQ Direct-Q®).

## Mono-enzymatic electromechanical measurements

The obtained Ppy/DBS films were properly cut in order to use them as bio-electrochemical actuators, with a total length of 1.1 cm, of which 3 mm correspond to the free-standing actuator part. To maximize the current density in the actuator section, the corresponding enzyme-covered extremity had a width of 5 mm, while the moving bare Ppy was only 1 mm large. The two different free-standing Ppy/DBS films, modified with Bilirubin oxidase (BOD) or Glucose oxidase (GOx), respectively, (Supplementary Methods) were fixed on an inert support in the center of a beaker. The self-induced bio-electromechanical measurements were carried out in a 0.3 M PBS solution ($NaH_2PO_4$, Sigma-Aldrich, ≥99.0%, and $Na_2HPO_4$, Sigma-Aldrich, ≥99.0%) at 25 °C (pH = 5), either saturated with $O_2$ or in the presence of a 1 mM glucose solution (Sigma-Aldrich, 99.5%). The partially reduced Ppy/DBS film was obtained by applying a reduction potential of −0.3 V vs Ag/AgCl for 30 s, after the electropolymerization.

## Bi-enzymatic pulse motion

The biochemical redox actuator was designed by immobilizing separately the two enzymes and their corresponding osmium-based redox polymers at each extremity and at opposite faces of an oxidized Ppy/DBS strip (1 cm × 1.5 cm) (Supplementary Methods). The spontaneous motion of the actuator was studied in a 0.3 M PBS solution at 25 °C (pH = 5) saturated with $O_2$ at different glucose concentrations of 1, 2.5, 5, 7.5 and 10 mM. The error bars were calculated according to the three sigma rule, referring to data within three standard deviations from the mean.

## Lateral global motion

The asymmetrically modified biochemical actuator was designed by immobilizing separately the corresponding enzymes and the appropriate redox polymers at each extremity and at opposite faces of an oxidized Ppy/DBS strip (0.5 cm × 1.5 cm). Prior to the enzyme immobilization, the Ppy/DBS film was asymmetrically overoxidized via bipolar electrochemistry in a 0.2 M $LiClO_4$ (Sigma-Aldrich, 99.9%) aqueous solution by applying an electric field of 4 V cm$^{-1}$ for a given time. Afterward, GOx and BOD were immobilized at the border between the still electroactive middle section of the PPy strip and the overoxidized and reduced extremities, respectively. The spontaneous lateral motion was studied in a 0.3 M PBS at 25 °C (pH = 5), 5 mM glucose solution, saturated with $O_2$. Video processing and tracking were performed in all cases with ImageJ software version 1.54.

## Reporting summary

Further information on research design is available in the Nature Portfolio Reporting Summary linked to this article.

## Data availability

The datasets generated and analyzed in the frame of the current study are available from the corresponding author on request.

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

## Acknowledgements

We are grateful for financial support by the European Research Council (ERC) under the European Union's Horizon 2020 research and innovation program (grant agreement n° 741251, ERC Advanced grant ELECTRA, A.K.). S.A. acknowledges financial support of Università degli Studi di Milano.

## Author contributions

S.A. performed the experiments, wrote and edited the manuscript. G.S. performed experiments, wrote and edited the manuscript and treated the data. S.B. synthesized the redox polymer and tested the BOD under heterogeneous conditions. S.G. produced, purified and tested the BOD in homogeneous solution. N.M. discussed the results and edited the manuscript. A.K. proposed the research project, designed the experiments, provided resources, and edited the manuscript.

## Competing interests

The authors declare no competing interests.
