## [Peer Review File · Nature Communications]

Bi-enzymatic chemo-mechanical feedback loop for continuous self-sustained actuation of conducting polymersREVIEWER COMMENTS

Reviewer #1 (Remarks to the Author):

The manuscript "Bi-enzymatic chemo-mechanical feedback loop for continuous self-sustained actuation of conducting polymers" shows a very interesting concept to acquire oxidation without external electric field or applied potential. Using chemical oxidants in solution has been done before (10.1016/j.snb.2009.06.044). There are some parts in the manuscript that needs clarity and other parts that need additional experimental before this work is suitable in Nature Communications.

1. The authors used PPy/DBS for conducting polymers and the reaction taking place in PBS buffer. PBS contains NaCl, KCl and other salts. Which cations moves in PPy/DBS at reduction? From older report (10.1016/S0167-2738(02)00530-1) in aqueous solution it is known that anions and cations move in PPy/DBS during redox. Therefore the authors took an assumption rather confirmation which ions led to volume change.

2. The PPy/DBS was deposited at quite high current density also seen the rough morphology at SEM images. The opposite SEM images looks like the side of working electrode deposition. A cross section image would give more clarity. Could it be that overoxidation taking place already at electropolymerization? Could the author provide the electrolymerization curve (supplementary)?

3. The thickness of PPy/DBS need to be considered as well hence the actuation performance depend on thickness of PPy/DBS (10.1117/12.776060, 10.1088/2399-7532/aae3e0). Please provide the thickness of deposited PPy/DBS and the conductivity (oxidized state) of such. In case overoxidation taking place during actuation such decrease in conductivity would confirm such.

4. It would be interesting if possible to verify how much charge is created in PPy/DBS during enzymatic reaction. Maybe using a comparison of chronamperometric measurements related to actuation response and those obtained over enzymatic reactions. In any case the bending displacement against applied frequency should be included.

5. The authors said that those bending occurred up to 10 min. This is a bit short to using such device in soft robotics. What are the reason for the decay of actuation? Is it the enzyme activity or other reasons? please clarify. Are there any creep taking place?

6. From electrochemical measurements (supplementary) why did the authors used LiClO₄ aq solution and not PBS buffer as shown in experiments? Can those different ions compared to explain the actuator mechanism?

7. For other experiments the authors used a hybrid actuator with one is PPy/DBS and the other PPy in LiClO₄ in ACN. Similar concepts have been studied before (10.1117/12.2009007, 10.1016/j.electacta.2016.02.104). There some parts not mentions that a full redox cycle can exchange cation/anions through the hybrid membrane and no further actuation are obtained. Therefore a certain potential range have to be used to achieve those hybrid actuator function.

Reviewer #2 (Remarks to the Author):

This paper describes a conducting polymer actuator fed by glucose and oxygen through enzymatic reactions that originate the electrons needed to produce the mechanical actuation in a continuous manner through an interesting chemo-mechanical feedback loop. Although conducting polymer actuators powered by enzymatic reactions have already been proposed in one of the already cited references, the chemo-mechanical feedback loop for continuous self-sustained actuation is clever and of great interest (this can in fact be highlighted in the introduction and discussion more clearly). The paper is in general, in my opinion, well written, but there is still room for improvement:

The paper is in general, in my opinion, well written, but there is still room for improvement:

1. References are missing in line 25 "artificial proteins". The rest of kinds of conducting polymers in the first sentence of the article are supported by proper references, however, artificial muscles based on artificial proteins are not supported by such reference(s).

2. In line 34, the authors claim that "CP actuators are known to exhibit a higher force and strain compared to natural muscles". Do the authors mean "stress" instead of "force"? Or could they clarify this aspect?

3. Figure 1E is hard to understand, could the authors clarify if the reactions happen in sequential order or not?
4. Figure 2A: Could the authors clarify why the electrodes "prepared by immobilization, separately, each enzyme with its corresponding redox mediator on the surface of a free-standing Ppy/DBS film" (lines 132-133) do not show the characteristics of only Ppy?
5. Figure 2A: what are the units of the normalized current?
6. Figure 2A: Why was the electrolyte used for the cyclic voltammogram of only Ppy different (0.1 M TBAP/CAN, line 164) than those of the enzymes immobilized ones (PBS)? How can they be compared?
7. Figure 2: Why are the experiments done at different temperatures (37.5 °C, line 162, and 25 °C, lines 166 and 170)?
8. Figure 2D: It is written that "Error bars correspond to three repetitions" (lines 171, 172). What kind of error is presented there?
9. The authors mention (lines 155, 156) that "the enzymatic reduction of oxygen leads to the oxidation of the partially reduced Ppy/DBS film". How was the initial redox state of the Ppy/DBS film achieved for every experiment?
10. Video S6: Could the authors clarify why are there such big steps between the frames and does not look a more continuous movement?
11. Methods:
 - o how were the glass slides coated with gold (line 446)? The thickness and quality of the gold may have an influence on the electrochemical experiments.
 - o What was the size of such gold-coated glass slides? The electropolymerization current is given, but the current density depends on the size of the electrodes.
 - o Mention in the main text that more details can be found in the supporting information would be good, especially on how were the enzymes and the osmium-based redox polymers immobilized on the Ppy/DBS.

Reply to referee comments on manuscript “Bi-enzymatic chemo-mechanical feedback loop for continuous self-sustained actuation of conducting polymers” (Manuscript number: NCOMMS-23-21282-T)

Reviewer: 1

The manuscript "Bi-enzymatic chemo-mechanical feedback loop for continuous self-sustained actuation of conducting polymers" shows a very interesting concept to acquire oxidation without external electric field or applied potential. Using chemical oxidants in solution has been done before (10.1016/j.snb.2009.06.044). There are some parts in the manuscript that needs clarity and other parts that need additional experimental before this work is suitable in Nature Communications.

We appreciate the referee’s positive judgment of our submission and address below point by point the more specific issues:

1. The authors used PPy/DBS for conducting polymers and the reaction taking place in PBS buffer. PBS contains NaCl, KCl and other salts. Which cations moves in PPyDBS at reduction? From older report (10.1016/S0167-2738(02)00530-1) in aqueous solution it is known that anions and cations move in PPy/DBS during redox. Therefore, the authors took an assumption rather confirmation which ions led to volume change.

We perfectly understand the referee’s concern. The ionic transport during the charge/discharge process of conducting polymers is a complex and well-known phenomenon that strongly depends on multiple parameters, i.e. type of material, internal organization of the matrix, applied potential or current, size and type of ion and thickness of the film. In this context it is well established that for thin-films (below 1 μm) an exchange of anions and cations can take place during the charge/discharge process. However, in this work we deposited polymer films with a thickness of around 70 μm which can entrap the doping ion used during the electropolymerization, in this specific case DBS. This led us to the conclusion that the volume change is associated with an exclusive cation exchange. In addition, we agree that the PBS buffer contains multiple salts with Na^+ and K^+ as main cations. In this context the key parameter for the expansion of the film is not the size of the ion itself, but the size of its hydration shell. Previous works have demonstrated that the most efficient volume change can be achieved with Li^+ in comparison with Na^+ or K^+ , due to the difference in their hydration shell (≈ 5.4 , ≈ 4.4 and ≈ 2.1 for Li^+ , Na^+ and K^+ respectively) (10.1002/anie.201709038 and 10.1007/s11581-009-0393-6). Thus, under the conditions used in this work, it is reasonable to assume that the observed actuation is mainly associated with the exchange of Na^+ present in the PBS buffer. This aspect is now discussed in the revised manuscript and the corresponding references were included.

2. The PPy/DBS was deposited at quite high current density also seen the rough morphology at SEM images. The opposite SEM images looks like the side of working electrode deposition. A cross section image would give more clarity. Could it be that overoxidation taking place already at electropolymerization? Could the author provide the electropolymerization curve (supplementary)?

We thank the referee for highlighting an important aspect of this work. The electropolymerization was carried out galvanostatically, thus we recorded the potential as a function of time. As it can be seen from the curve, the potential remains relatively constant, in a range between 0.65 and 0.7 V vs Ag/AgCl, and this value is significantly lower than the overoxidation potential of the monomer and of its corresponding polymer. The optical picture of the cross section of the film allows visualizing the different morphologies of the two faces and an evaluation the thickness of the polymer ($\approx 70 \mu\text{m}$). The potential transient plot

obtained during the electropolymerization and the picture of the cross-section of the film have been now included in the new version of the manuscript.

3. The thickness of PPy/DBS need to be considered as well hence the actuation performance depend on thickness of PPy/DBS (10.1117/12-776060, 10.1088/2399-7532/aae3e0). Please provide the thickness of deposited PPy/DBS and the conductivity (oxidized state) of such. In case overoxidation taking place during actuation such decrease in conductivity would confirm such.

We appreciate the referee's request. The thickness of the electro-synthesized polymer was evaluated to be around 70 μm , whereas the conductivity can be estimated by taking into account the geometric parameters of the film (1 cm x 1.5 cm) and its resistance (380 Ω), obtaining a value of 0.56 S cm^{-1} . We perfectly agree with the referee that both parameters are of fundamental importance for the actuation performance of the polymer. Higher conductivity values facilitate the electron exchange from the anodic to the cathodic side of the device, whereas thin films favor the ion exchange improving the mechanical bending. However, thin polymers suffer from a lack of mechanical stability and high conductivities leading to charge trapping limitations. The electropolymerization conditions used in this work have been optimized previously to lead to the most efficient bending during the redox processes. In addition, these conditions, as stated in our answer to question 2, avoid the overoxidation of the monomer and of the obtained polymer, which provides better chemical, electrochemical and mechanical stability. Thickness and conductivity values were indicated in the revised version of the manuscript, and a brief paragraph about the influence of the thickness and conductivity on the actuation performance was included in the discussion section. Concerning the possible overoxidation of the film during the enzymatic redox transfer, it can be seen in Figure 2a that both redox processes associated with the enzymatic reactions are confined to a region where the Ppy is electroactive and below the overoxidation potential. This aspect was now clarified in the new version of the manuscript.

4. It would be interesting if possible to verify how much charge is created in PPy/DBS during enzymatic reaction. Maybe using a comparison of chronoamperometric measurements related to actuation response and those obtained over enzymatic reactions. In any case the bending displacement against applied frequency should be included.

We agree with the referee that it might be interesting to provide quantitative values of the charge produced during the enzymatic reactions. However, the suggested comparison is not possible since the areas used for the mentioned experiments are different. In addition, the conductivity of the Ppy substrate needs to be considered in order to estimate the charge produced within the polymer matrix via the coupled enzymatic reactions. A deeper study could be carried out by evaluating separately the current produced by the GOx/Ppy or Ppy/BOD systems.

Concerning the last point, we are not sure to understand the referee's request. We cannot provide a correlation between the bending displacement and the applied frequency, since in this work we are not applying a frequency. It is the frequency of the spontaneous oscillation. Perhaps the referee is referring to the amplitude as a function of time, however, due to the frame rate of the videos it is not possible to provide a more detailed analysis of the bending. In order to avoid a possible confusion with an externally applied frequency we replaced everywhere in the manuscript "pulse frequency" by "oscillation frequency".

5. The authors said that those bending occurred up to 10 min. This is a bit short to using such device in soft robotics. What are the reason for the decay of actuation? Is it the enzyme activity or other reasons? please clarify. Is there any creep taking place?

We thank the referee for his/her observation. Indeed, the decay of the actuation at longer times is associated mainly with two factors. First, a leakage of the enzyme from the surface of the device, and second, an intrinsic decay of the enzymatic activity. Both phenomena lead to less activity in terms of electron transfer, which translates into a decay of the actuation. However, both aspects might be optimized in future work, because it is possible to cover the enzyme layers with protecting membranes and thus avoiding the loss/leaking of enzymes due to the mechanical stress related to the local hydrodynamics and shear forces present during actuation. Furthermore, it is possible to use genetically modified enzymes which have a much longer lifetime than the enzymes used in this proof-of-concept study. We mention these aspects now in the discussion section of the manuscript.

6. From electrochemical measurements (supplementary) why did the authors use LiClO₄ aq solution and not PBS buffer as shown in experiments? Can those different ions be compared to explain the actuator mechanism?

We understand this justified question of the referee. As stated above in question 1, the main feature causing the expansion of the film is the size of the hydration shell of the cation. Thus, the actuation during the bi-enzymatic chemo-mechanical feedback loop is caused by the exchange of Na⁺, whereas, for the conventional experiments the actuation is based on an uptake/release of Li⁺. However, both cations present relatively similar hydration shells (≈5.4 and ≈4.4 for Li and Na, respectively), thus in a first-order approximation the electromechanical motion associated with both cations can be compared. The conventional actuation measurements were carried out in order to corroborate the difference in the ion uptake and release kinetics at the rough and smooth face of the Ppy film during the charging/discharging process. Due to the relatively similar hydration shell of Li⁺ and Na⁺ it is expected that in both experiments the angular speed presents a similar tendency.

7. For other experiments the authors used a hybrid actuator with one is PPy/DBS and the other PPy in LiClO₄ in ACN. Similar concepts have been studied before (10.1117/12.2009007, 10.1016/j.electacta.2016.02.104). There some parts not mention that a full redox cycle can exchange cation/anions through the hybrid membrane and no further actuation are obtained. Therefore, a certain potential range have to be used to achieve those hybrid actuator function.

We are not completely sure that we have understood the referee's request. Indeed, the potentiodynamic behavior of the redox mediators was studied by using a Ppy/DBS electrode, whereas the electrochemical characterization of a bare Ppy film was carried out in a LiClO₄/ACN solution. The latter was performed under these conditions in order to obtain a well-defined electroactive window exhibiting the main redox processes of Ppy. We agree with the referee, that during the charging/discharging process of the Ppy film in ACN, it is possible to exchange cations and anions, however, once again this potentiodynamic study was used only to indicate that the electrochemical activities of the redox mediators are located in a region of potential where the Ppy film is electroactive. All the electromechanical studies were carried out with a Ppy/DBS film which, as stated above in question 1, can mainly exchange cations. Thus, the system is not similar to the asymmetric bilayer actuators proposed by Otero et al.

Reviewer: 2

This paper describes a conducting polymer actuator fed by glucose and oxygen through enzymatic reactions that originate the electrons needed to produce the mechanical actuation in a continuous manner through an interesting chemo-mechanical feedback loop. Although conducting polymer actuators powered by enzymatic reactions have already been proposed in one of the already cited references, the chemo-

mechanical feedback loop for continuous self-sustained actuation is clever and of great interest (this can in fact be highlighted in the introduction and discussion more clearly).

The paper is in general, in my opinion, well written, but there is still room for improvement:

We thank the referee for his/her very positive judgment of our contribution and the recommendation for publication. We address the different specific points in the following

1. References are missing in line 25 “artificial proteins”. The rest of kinds of conducting polymers in the first sentence of the article are supported by proper references, however, artificial muscles based on artificial proteins are not supported by such reference(s).

We apologize for these missing references. Two new references have been now included in the revised version of the manuscript.

2. In line 34, the authors claim that “CP actuators are known to exhibit a higher force and strain compared to natural muscles”. Do the authors mean “stress” instead of “force”? Or could they clarify this aspect?

We thank the referee for pointing out the mistake, this has now been modified.

3. Figure 1E is hard to understand, could the authors clarify if the reactions happen in sequential order or not?

We are sorry that the meaning and interpretation of Figure 1E was not clear enough. Actually, the arrow in the figure was supposed to illustrate that the coupled redox reactions follow a thermodynamic chain which is an overall downhill reaction. It was not supposed to be a timeline. The sequential character of the different events is indicated in Figure 1A-D. We have clarified this now in the new version of the manuscript.

4. Figure 2A: Could the authors clarify why the electrodes “prepared by immobilization, separately, each enzyme with its corresponding redox mediator on the surface of a free-standing Ppy/DBS film” (lines 132-133) do not show the characteristics of only Ppy?

We appreciate this interesting observation. In fact, in this case, the free-standing Ppy acts only as a conductor, thus no redox reactions involving the PPy take place. The only faradaic current is the one associated with the oxidation/reduction of the redox mediators. However, in both cases, it is possible to appreciate the capacitive current associated with the Ppy film. This has been indicated now in the new version of the manuscript.

5. Figure 2A: what are the units of the normalized current?

We apologize this was not clear enough in the previous version. The normalized current does not have units, since for each potential the ratio between the current and the maximum current was evaluated.

6. Figure 2A: Why was the electrolyte used for the cyclic voltammogram of only Ppy different (0.1 M TBAP/CAN, line 164) than those of the enzymes immobilized ones (PBS)? How can they be compared?

We agree with the referee that this can be misleading. Indeed, the potentiodynamic behavior of the bare Ppy film was carried out in a LiClO₄/ACN solution in order to obtain a well-defined electroactive window exhibiting the main redox processes of Ppy. We agree that it is not possible to make a direct comparison between these measurements and the potentiodynamic behavior of the redox mediators in PBS. However, this potentiodynamic study was used only to indicate that the electrochemical reactions of the redox mediators are found within a region of potential where the Ppy film is electroactive.

7. Figure 2: Why are the experiments done at different temperatures (37.5 °C, line 162, and 25 °C, lines 166 and 170)?

The potentiodynamic behavior of the redox mediators was evaluated at 37.5°C, since this is the optimized temperature where the enzyme exhibits the highest activity. The dynamic behavior of the polymer actuator was studied at 25 °C in order to avoid thermal convection, which can influence the motion.

8. Figure 2D: It is written that “Error bars correspond to three repetitions” (lines 171, 172). What kind of error is presented there?

The error bars were calculated according to the three sigma rule referring to data within three standard deviations from the mean.

9. The authors mention (lines 155, 156) that “the enzymatic reduction of oxygen leads to the oxidation of the partially reduced Ppy/DBS film”. How was the initial redox state of the Ppy/DBS film achieved for every experiment?

The pristine Ppy/DBS film obtained after galvanostatic electropolymerization is in a charged state with a conductivity of 0.56 S cm⁻¹. However, for the experiment pointed out by the referee, the partially reduced state of the Ppy/DBS film was obtained by applying a reduction potential (-0.3 V vs Ag/AgCl for 30 seconds) after the electropolymerization. This has been clarified now in the experimental section of the revised manuscript.

10. Video S6: Could the authors clarify why are there such big steps between the frames and does not look a more continuous movement?

We thank the referee for this interesting observation. Such big steps of the actuator are not related to the resolution of the video, but originate from the pulse mechanism shown in Video S4. As it can be seen, the actuator exhibits a unidirectional “jumping” motion, due to the relatively small electroactive region of Ppy. As described in the manuscript, a difference in the ion exchange kinetics, due to the surface anisotropy of the material, induces a continuous pulse process in the small electroactive middle part of the polymer causing this pulsed motion. Since Video S6 presents the top view of the motion, the full pulse or “jumping” process cannot be properly appreciated. This was now specified in the new version of the manuscript.

11. Methods:

o how were the glass slides coated with gold (line 446)? The thickness and quality of the gold may have an influence on the electrochemical experiments.

We apologize for the lack of clarity. For this work we used commercial gold-coated glass slides with a rather smooth surface and a defined thickness between 300 and 400 nm. We agree that the thickness and quality of the gold may affect the electropolymerization however the synthesis of polypyrrole has been optimized previously and the methodology is well defined. We specify in the new version of the manuscript the use of commercial gold-coated glass slides.

o What was the size of such gold-coated glass slides? The electropolymerization current is given, but the current density depends on the size of the electrodes.

We agree with the referee’s important request. The electrode area, where the electropolymerization is carried out, is 2 cm², thus a current density of approximately 2 mA cm⁻² was used. However, the most important aspect of the galvanostatic method is the efficient control of the potential which, under the

conditions of this work, remains between 0.65 and 0.7 V vs Ag/AgCl. This allows avoiding parasitic reactions that can damage the polymer, such as overoxidation. The potential transient obtained during the electropolymerization was included in the new version of the supporting information.

o Mention in the main text that more details can be found in the supporting information would be good, especially on how were the enzymes and the osmium-based redox polymers immobilized on the Ppy/DBS.

The related modifications have been made.

REVIEWERS' COMMENTS

Reviewer #1 (Remarks to the Author):

The authors made revision and answered all open question. They add additional explanations. The manuscript now in publishable form

Reviewer #2 (Remarks to the Author):

I appreciate the responses to my comments from the authors. They have successfully addressed most of my concerns about the manuscript. However, most of them have not been included in the main manuscript nor in the supporting information. Such discussions/clarifications would be useful for any reader of the paper and could be included in the text (see questions 4, 5, 6, 7, 8, and also 9 that sorry, but could not find it marked in yellow in the text).

Also, it would be beneficial to include the supplier (and even reference) of the Au-coated glass slides (regarding question 11), and discussion concerning question 4 on why/how the PPy is considered only a conductor when in those conditions the PPy can be also oxidized/reduced itself as shown in figure 2a (although in other electrolyte, as mentioned in question 6).

Reply to referee comments on manuscript “Bi-enzymatic chemo-mechanical feedback loop for continuous self-sustained actuation of conducting polymers” (Manuscript number: NCOMMS-23-21282-T)

Reviewer: 2

I appreciate the responses to my comments from the authors. They have successfully addressed most of my concerns about the manuscript. However, most of them have not been included in the main manuscript nor in the supporting information. Such discussions/clarifications would be useful for any reader of the paper and could be included in the text (see questions 4, 5, 6, 7, 8, and also 9 that sorry, but could not find it marked in yellow in the text).

Also, it would be beneficial to include the supplier (and even reference) of the Au-coated glass slides (regarding question 11), and discussion concerning question 4 on why/how the PPy is considered only a conductor when in those conditions the PPy can be also oxidized/reduced itself as shown in figure 2a (although in other electrolyte, as mentioned in question 6).

We appreciate the referee’s positive judgment of our submission and we apologize for the missing more detailed discussion in the manuscript of some aspects that were mentioned in the previous reply letter. The comments pointed out by the referee are now included in the new version of the manuscript.